# Increasing Condom Use and STI Testing: Creating a Behaviourally Informed Sexual Healthcare Campaign Using the COM-B Model of Behaviour Change

**DOI:** 10.3390/bs12040108

**Published:** 2022-04-15

**Authors:** Sara Bru Garcia, Małgorzata Chałupnik, Katy Irving, Mark Haselgrove

**Affiliations:** 1School of Psychology, University of Nottingham, Nottingham NG7 2RD, UK; mark.haselgrove@nottingham.ac.uk; 2School of English, University of Nottingham, Nottingham NG7 2RD, UK; malgorzata.chalupnik@nottingham.ac.uk; 3Healthcare Research Worldwide, High St. 46, Wallingford OX10 0DB, UK; k.irving@hrwhealthcare.com

**Keywords:** COM-B model, behaviour change, co-creation, sexual health, public health campaigns

## Abstract

Sexually transmitted infections (STIs) are a major public health challenge. Although theoretically informed public health campaigns are more effective for changing behaviour, there is little evidence of their use when campaigns are commissioned to the commercial sector. This study describes the implementation of the COM-B model to a sexual health campaign that brought together expertise from academics, sexual healthcare, and marketing and creative professionals. Insights were gathered following a review of the relevant academic literature. Barriers and facilitators to condom use and STI testing were explored with the use of the COM-B model and the Behaviour Change Wheel in a workshop attended by academics, behavioural scientists, healthcare experts and creative designers. Feedback on the creative execution of the campaign was obtained from healthcare experts and via surveys. Barriers to psychological capability, automatic and reflective motivation, and social opportunity were identified as targets for the campaign, and creative solutions to these barriers were collaboratively devised. The final sexual health campaign was rated positively in its ability to change attitudes and intentions regarding the use of condoms and STI testing. This study describes the implementation of the COM-B model of behaviour change to a public sexual health campaign that brought together academics, public and commercial sector expertise. The barriers and facilitators identified in this collaborative process represent potential targets for future public health communication campaigns.

## 1. Introduction

Sexually transmitted infections (STIs) are a major public health challenge with significant economic, social, and psychological implications [1,2]. According to recent data from Public Health England’s STI surveillance system [3], the reporting of new cases of STIs was steadily on the increase up until 2019. The decreased number of new diagnoses in 2020 may not indicate a lower rate of transmission of STIs but the disruption to some sexual health services (SHS) caused by COVID-19, declared a pandemic by the World Health Organization in March 2020. The large numbers of diagnoses observed, despite the constrained capacity for STI testing, combined with the results of community surveys instead reveal the sustained transmission of STIs [3]. In this context, the focus on public health messages around sexual health to facilitate the uptake of methods of STI prevention and diagnosis, such as condom use and STI testing, remains a high priority. These aspects of public health messages around sexual health, both of which are concerned with behaviour change, are the key focus of this article. Specifically, we consider how the COM-B model of behaviour change can be implemented in the design of a sexual health campaign used in a public space.


**Behaviour Change Model**


There are a number of theoretical models to understand processes of behaviour change. The COM-B model was chosen for the purpose of developing the sexual health campaign presented in this paper because it integrates core theoretical and empirical constructs present in early theories of behaviour change [4]. Additionally, the COM-B model can be presented at different levels of complexity based on the psychological knowledge of the audience, which made it particularly useful for a co-creation session that included sexual healthcare experts from the UK National Health Service, academic psychologists from a UK university, and behavioural scientists and creative professionals from marketing and creative agencies. The model describes a behaviour system where *capability*, *opportunity* and *motivation* need to be present at sufficient levels for a behaviour, such as using a condom, to occur. Capability can be either physical or psychological; opportunity can be either social or environmental; and motivation either automatic or reflective. These three components interact and influence each other, forming behaviour systems that can vary in complexity. To facilitate moving from understanding a behaviour to changing it, each COM-B component is linked to the intervention function most likely to influence them via the Behaviour Change Wheel (BCW) [4]. For example, the “persuasion” function might be effective for influencing motivation, but it is not likely to influence “capability”. Any given intervention might change one or more components in the behaviour system. For instance, the “training” function is likely to influence capability but also motivation, see Table 1.

The COM-B model has predominantly been employed to identify barriers and enablers to condom use and STI testing, with a focus on digital interventions [1,5,6,7]. Digital interventions offer a controlled environment to implement and evaluate COM-B. However, public health campaigns are often designed to be displayed in public spaces, where the competition for people’s attention is high and the attitudes and beliefs about the target health behaviours are varied. To address this challenge, the current study outlines the development of the “*Come together safely this summer*” public health campaign, commissioned by UK county councils and collaboratively developed by behavioural psychologists and scientists, sexual health experts and communication and marketing experts.


**Sexual health and the application of the COM-B model**


The COM-B model has been successfully applied to a variety of health behaviours at both individual and organisational levels [8,9,10,11]. Prior research has predominantly focused on the application of COM-B and the BCW to identify barriers and facilitators regarding the use of condoms and STI testing [5,7,12]. For example, researchers have identified the perception of diminished pleasure [5,13] or the fear of what peers might think of the person who carries or suggests the use of a condom [1,7] as common barriers to their use.

A focus of existing research has been to apply COM-B and the BCW to encourage condom use in interactive digital interventions, such as the MenSS website [5]. Overall, systematic reviews of individual-level behaviour change interventions have demonstrated their effectiveness in reducing key infection-risk-related behaviours, although more research is needed to determine the generalisability of these results to the UK setting [14]. Whilst evidence shows that public health campaigns based on previous research and evidence are more effective at changing behaviour [1], there is a need for more research about how the COM-B model and other relevant findings can be integrated into health campaigns commissioned for display in urban public settings. We provide a description of a collaborative approach that capitalises on expertise from academic psychologists from a UK university, and behavioural science experts whilst engaging critical stakeholders and commercial sector expertise.


**Aims and Objectives**


This study explores how barriers and facilitators to condom use and STI testing can be addressed in public health messages around sexual health, specifically, how these campaigns can be informed by the COM-B model of behaviour change. We demonstrate the value of a collaborative process that combines academic, public and commercial sector expertise to design appealing public health campaigns that are effective for changing behaviour. 

## 2. Materials and Methods

The implementation of the COM-B model of behaviour change to the campaign consisted of three key stages. The first stage involved a review of the relevant academic literature on condom use and STI testing in previous successful interventions. The second stage focused on a COM-B workshop with relevant experts. The final stage involved gathering feedback on two provisional creative executions, initially from healthcare experts and later from the target population via a survey. The COM-B model and BCW were used as frameworks to guide the overall process. Data were collected and securely stored by the Together Agency. 


**Data collection**

**Academic literature**


A targeted review of academic literature identified relevant insights on condom use and STI testing in previous successful interventions and campaigns. Search terms included ‘*condom*’, ‘*intervention*’, ‘*behavioural*’, ‘*risk perception*’ and ‘*barriers*’. The articles selected included insights on framing and communication of risk, perceived barriers of condom use and behavioural domains key to condom use and STI testing. The final sample of studies selected [5,13,15,16,17,18,19,20] were summarised, and their main findings prepared for presentation to key stakeholders, who in this case were Derbyshire Community Health Services NHS Foundation Trust.


**Expert COM-B workshop**


A three-hour long workshop to inform the design of the campaign was held. Attendees to the workshop represented all the areas of expertise included in this collaborative process: academics from the schools of Psychology and English at a UK university, behavioural science experts from a private-sector marketing company, creative experts from a branding and marketing company and the clients: healthcare professionals, including clinicians and commissioners from Derbyshire Community Health Services NHS Foundation Trust (UK). Participants were presented with the key findings from the literature review and asked to prioritise target behaviours. After specifying the target behaviours to be promoted in the campaign, all attendees participated in the development process by identifying barriers and facilitators to peoples’ capability, opportunity and motivation to use condoms and be tested for STIs. This co-creation process allowed for the identification of relevant intervention functions and guided the decisions regarding the creative execution of the campaign. For example, the normalisation of condom-related behaviours (i.e., using condoms or carrying condoms) was deemed to be a high priority to target barriers to safer sex. This was addressed both visually and through the use of behaviourally informed messaging.


**Feedback on creative design**


The outputs from the COM-B workshop served as inputs for a team of creative designers and copywriters from the Together Agency. Their brief was to translate the identified behavioural barriers and facilitators to condom use and STI testing into a campaign that was able to engage the target audience. As a result of this process, two provisional campaigns were executed: “*Come together safely this summer*” and “*Don’t blow it this summer*”.

These two resulting creative executions were assessed, first with feedback from the sexual health specialists and, later, with a survey designed to measure key attitudes and reactions to these behaviourally informed campaigns. The survey was circulated across the target population: heterosexual and LGBTQ+ people aged 16 in the county and neighbouring areas. 

The final version of the survey was agreed with the clients. The survey was advertised on the county’s Community Health Services website and received responses from 48 volunteers (33 women, 12 men and 3 respondents who preferred not to disclose their gender). The survey was intended to aid decision making and refine the creative execution of the final health campaign.


**Survey statistical analysis**


The survey was administered in the online platform SurveyMonkey. Participants’ data were analysed using IBM SPSS, version 26 [21]. Where rating scales were used they ranged from strongly disagree to strongly agree on a 1–5 Likert scale. To determine if one of the versions of the campaign was preferred over the other one, a paired sample t-test was conducted. Some critical questions designed to measure attitudes and intentions (e.g., This advert could change my attitudes towards this health issue) were analysed using a one-sample t-test to determine if the ratings were significantly different from neutral (rating level 3); that is, whether our behaviourally informed campaign had a significant effect on people’s attitudes and intentions towards condom use and STI testing. A level of significance of *p* < 0.05 was adopted for all statistical tests.

## 3. Results


**Expert COM-B workshop**


Of the range of healthcare experts who attended the workshop, only some of them were familiar with the COM-B model of behaviour change. Even fewer had considered the barriers and facilitators to condom use and STI testing in terms of capability, opportunity and motivation to enact the behaviour. However, all attendees reacted positively towards the co-creation process and towards applying a well-established model of behaviour change to this public health campaign. The workshop facilitator probed attendees to think about the challenges that the target population faced regarding safe sex and STI testing. For the full mapping of COM-B components, corresponding BCW intervention functions and ideas for creatively targeting barriers, please see Table 1.


**Capability: physical and psychological**


Within the *capability* component of COM-B, attendees described both physical and psychological capabilities (and limitations) to safe sex and STI testing. For example, most participants acknowledged the need for further educational and training efforts to ensure young people in particular had the skills to use a condom correctly, but also the skills to navigate conversations and effectively negotiate the use of contraception with sexual partners. This was captured by several comments from the healthcare experts attending the session, who said “*I don’t think we can assume the information that they’ve had, when they’ve had it, how much they’ve understood it”* and that “*There’s definitely something in there about the younger population being able to negotiate those conversations (…) negotiating safer sex (…) help instigate that conversation (…) probably is the same for older populations*”.

A major issue that participants highlighted was the lack of general knowledge about condom types and ease of access. The consensus among healthcare experts was that young people lack knowledge about “*where to get them* [condoms]*. Do they know about the C-card scheme, do they know about being able to get them from pharmacies and GPs? Being able to get your hands on them to begin with*”. This lack of knowledge was identified as a barrier to dispel some of the myths surrounding condom use and pleasure. For example, one health practitioner mentioned how “*using condoms reduces pleasure is a popular thought*”, even though this might simply be a consequence of the user not having found the right condom. 

These barriers were tackled by the creative team by using behaviourally informed messaging that served an educational purpose. Specifically, the messages highlighted the variety of condoms available to users and the different ways in which they could access condoms, including free condoms, targeted to the different age groups. 


**Opportunity: physical and social**


Although the *opportunity* component of COM-B was arguably more difficult to influence with a purely communication-based campaign (specifically, the physical opportunity subcomponent), the social subcomponent within *opportunity* was identified as particularly relevant to encourage safe sex and STI testing. All attendees identified the normalisation of sex as a crucial aspect of the campaign. This was captured by multiple comments, all of which emphasised how “*Sex is normal. However, you like to do it is normal*” and “*Is a key message, is normal, and therefore enjoy it (…) yeah is that clear message about normalising it which is really key*”. As part of the idea of normalising sex and sex-related behaviours, some healthcare experts referred to the use of condoms and STI testing as embedded within the usual steps that anyone should be taking as part of their healthcare, with one attendee saying that we should be “*Normalising condom use and testing as being part of the normal everyday activities that loads and loads of people do, use and do*”.

The campaign addressed barriers to social opportunity by ensuring the visual representation of a wide range of identities and demographics and by encouraging the idea of condom use and STI testing as part of a holistic approach to healthcare and wellbeing.


**Motivation: reflective and automatic**


Both the reflective and automatic subcomponents within *motivation* were identified as essential influences to encourage safe sex and STI testing. Within the reflective subcomponent of motivation, multiple attendees pointed at the beliefs surrounding the consequences of using condoms as a powerful incentive “*simply yours and your partner’s health, safety and wellbeing, you’d hope that’s a big incentive* [to use condoms]”. Additionally, attendees highlighted the importance of framing these beliefs about the consequences of using condoms from a sex-positive perspective. Instead of focusing on factual information stating that condoms are an effective way to prevent STIs and undesired pregnancies, focusing on the positive emotions associated with condom use can be an enjoyable part of a sexual encounter, whilst giving you peace of mind and allowing you to focus on your own and your partner’s pleasure. In the words of one of the healthcare practitioners “*Most of the young people I come into contact with seem to accept them as a fact of life (…) most young people are uninspired by them (…) it’s a question of inspiring some enthusiasm and sparking something in the conversation really (…) the pleasure that you can get from using condoms, rather than the need to use condoms (…) it’s about making sure that you have the right condom for you (…) actually there is a lot of pleasure to be had in using condoms*”. 

Workshop attendees also identified barriers to safe sex within the automatic subcomponent of *motivation*. The role of emotions, urges and (lack of) habits influencing decision making around safe sex resonated in the views shared by most attendees. For example, simply keeping condoms in places where sexual relationships tend to happen was identified as a key habit to facilitate their use, as a healthcare practitioner illustrated “*I usually say it’s no good having a condom in your bag at the bottom of the stairs when you’re about to have sex with your partner in the bedroom*”. 

These barriers were addressed creatively in different ways. The messaging was optimised to be persuasive, rather than purely informative, with the use of a positive frame and a focus on the positives of condom use. Additionally, both condoms and STI testing were shown as embedded within anyone’s routine. For example, condoms were shown side-by-side with other everyday items as a ‘going out checklist’, encouraging the target audience to see condom use and condom carrying as embedded within a habit—take your keys, mask, phone and condom with you before going out. 


**Post-workshop survey: key insights**


Participants’ demographics are provided in Table 2. Forty-eight participants were recruited to assess the two health campaigns that resulted from the process of combining COM-B and creative expertise. All participants gave informed consent prior to the start of the survey.

First, the two campaigns were compared for their relevance, interest, ease to understand and attention-catching properties. A paired sample t-test confirmed that the two campaigns did not differ significantly *t*(47) = 2.01, *p* > 0.05, with the mean rating for the “*Come together safely*” campaign (*M* = 3.44) only slightly lower than that for the “*Don’t blow it*” one (*M* = 3.48). Because the purpose of this survey was to aid decision makers in selecting the final campaign, we looked at the most critical items within the survey: those intended to measure intentions to change attitudes and target behaviours. Specifically, we focused on three questions:This advert would change attitudes towards the health issueThis advert would make me book an STI testThis advert would make me buy condoms

A one sample t-test, comparing participants’ ratings with neutral ratings (3) in our Likert scale, confirmed that both campaigns were rated as likely to change attitudes towards the health issue addressed (condom use or STI testing): *t*(47) = 2.98, *p* < 0.001 and *t*(47) = 4.06, *p* < 0.001 for the “*Come together safely*” and “*Don’t blow it*” campaigns, respectively. This reflects that both campaigns, which were directly informed by the insights obtained from the COM-B workshop, were seen as successfully influencing people’s attitudes towards the target behaviours. Ultimately, the decision to select the “*Come together safely this summer*” was taken after presenting the survey data to the clients and was driven by a self-reported increased intention to perform the target behaviours in the “*Come together safely this summer*” campaign, as shown in Figure 1. This campaign was also deemed to better reflect the positive framing of the campaign.

## 4. Discussion

This study presented a methodology that embedded academic evidence and the COM-B model into the processes and creative executions of a commercial marketing and communications agency to guide and inform the execution of a public health campaign designed to change behaviour effectively: “*Come together safely this summer*”. To achieve this, we selected key insights from an academic literature review and ran a COM-B workshop that explored the factors influencing people’s capability, opportunity and motivation to use condoms and get tested for STIs. We found that although most of the stakeholders involved were not familiar with the COM-B model, they all reacted very positively to this approach and to the aim of exploring and understanding barriers and facilitators to target behaviours, ensuring that the campaign was leveraging the right insights.

The study findings revealed that knowledge was a key barrier to target within psychological capability, with healthcare experts agreeing on the need to reinforce information, such as where to get condoms for free or the fact that condoms come in different sizes. The importance of social norms surrounding condom use and STI testing was also identified as key within the social opportunity subcomponent of COM-B. In particular, healthcare experts highlighted the need to approach sexual health behaviours just like any other health behaviour, encouraging the narrative of sexual healthcare as part of holistic healthcare and wellbeing. Within motivation, healthcare experts identified barriers in both the reflective and automatic subcomponents. They emphasised the need to encourage-positive emotions and sex-positive beliefs about the consequences of condom use, focusing on condoms as a means to relax and fully focus on yours and your partner’s pleasure, as opposed to focusing on condoms as a means to prevent STIs and unintended pregnancies. Habits were also identified as key targets of this campaign, with the idea of integrating carrying a condom as part of someone’s routine.


**Implications for practice and research**


Campaigns to increase the use of condoms and raise awareness about STIs are common in commercial and public contexts. However, the extent to which public health campaigns have applied rigorous and theoretically informed methods is unclear. A key strength of this research is that it was successful in putting together expertise from the academic and both the public and commercial sectors, capitalising on the knowledge of clinicians and practitioners, the creative skills of the Together Agency and the theoretical expertise of academics from a UK university and behavioural scientists in HRW. This has been identified as necessary in order to create very effective behaviour change campaigns [1]. The robustness of the methods and theoretically informed approach are likely to increase the effectiveness of creative outputs. Furthermore, understanding how barriers to capability, opportunity and motivation affect these target behaviours is critical to optimising future campaigns and communications.


**Limitations**


A limitation of the current study is that it did not capture information on immigration, the socio-economic status or ethnicity of the participants involved. Whilst individual-level behaviour change interventions were shown to be effective against infection risk-related behaviours, it must be acknowledged that broader structural factors, such as inequality, have an influence on sexual health behaviour. Capturing these data would be important to create more effective targeted campaigns and services in the future.

A second limitation of this study is that the sample of participants rating the two creative outputs were self-selected and mostly heterosexual women. Furthermore, it is possible that respondents who engaged with the survey, who would have been browsing the sexual health services website, already had favourable attitudes towards the use of condoms and STI testing. This could have been addressed by engaging users throughout the campaign development process, instead of engaging them only after development. Additionally, given how men often endorse more barriers to condom use, especially on the impact on pleasure factor [22], the sample would have benefited from more input from men.

## 5. Conclusions

The current study described the implementation of the COM-B model to a public sexual health campaign that brought together academia, the public and commercial sector expertise. The findings from the present study identified barriers and facilitators to condom use and STI testing and potential intervention functions to address these. These insights were translated into a creative campaign designed to be displayed across public spaces in a UK city. As the value of theoretically informed sexual health campaigns becomes more clear, we need to encourage collaboration between experts from different fields to effectively prepare for behaviour change.

## Figures and Tables

**Figure 1 behavsci-12-00108-f001:**
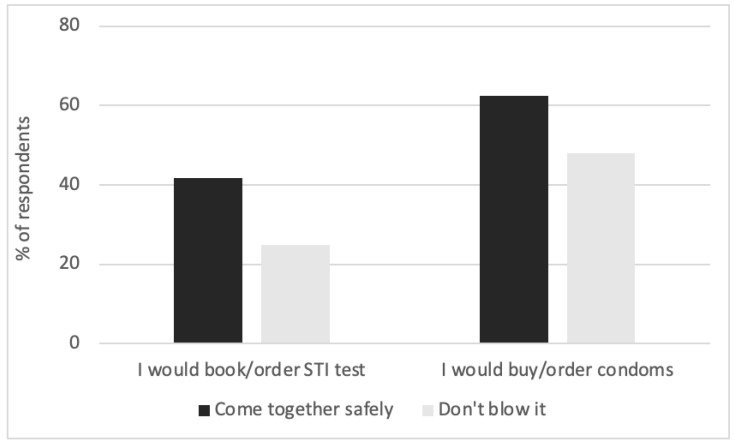
Percentage of respondents who reported an intention to perform the target behaviours—either buying or ordering condoms or STI tests—after seeing each campaign.

**Table 1 behavsci-12-00108-t001:** Relationship between the COM-B components and the BCW intervention functions. Crosses (×) indicate intervention functions that are more likely to have an influence on COB-B components.

COM-B Component	Intervention Functions								
	Education	Persuasion	Incentivisation	Coercion	Training	Restriction	Environmental Restructuring	Modelling	Enablement
**Physical Capability**					**×**				**×**
**Psychological Capability**	**×**				**×**				**×**
**Physical Opportunity**					**×**	**×**	**×**		**×**
**Social Opportunity**						**×**	**×**	**×**	**×**
**Automatic Motivation**		**×**	**×**	**×**	**×**		**×**	**×**	**×**
**Reflective Motivation**	**×**	**×**	**×**	**×**					

**Table 2 behavsci-12-00108-t002:** Survey participant demographics.

**Total Participants**	**48**
**Gender**	
Female	33 (68.75%)
Male	12
Prefer not to say	3
**Age**	
35+	21 (43.75%)
22–35	25
16–21	1
Prefer not to say	1
**Sexual orientation**	
Heterosexual or straight	38 (79.16%)
Homosexual	5
Bisexual	1
Other	1
Prefer not to say	3

## Data Availability

The datasets generated and/or analysed during the current study are not publicly available due to the involvement of two commercial partners in this project, but are available from the corresponding author on reasonable request.

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
