# Peer review of "Increasing Condom Use and STI Testing: Creating a Behaviourally Informed Sexual Healthcare Campaign Using the COM-B Model of Behaviour Change"

_behavsci, 2022, doi:10.3390/bs12040108_

Round 1
Reviewer 1 Report
Introduction
- line 42, During the Covid-19 pandemic, the general population was subjected to a greater stress load, with the appearance of anxiety disorders, reduced work performance, reduced night rest. please cite doi:10.7416/ai.2021.2439
- line 58, Men who have sex with men (MSM) experience significant inequalities in health and well-being. They are the group in the UK at the highest risk of acquiring a human immunodeficiency virus (HIV) infection. Guidance relating to both HIV infection prevention, in general, and individual-level behavior change interventions, in particular, is very necessary. An evidence synthesis of the clinical effectiveness of behavior change interventions to reduce risky sexual behavior among MSM after a negative HIV infection test. To identify effective components within interventions in reducing HIV risk-related behaviors and develop a candidate intervention. To host expert events addressing the implementation and optimization of a candidate intervention. An interesting systematic review of the clinical effectiveness of individual behavior change interventions was conducted. The analysis demonstrated that individual-level behavior change interventions are effective in reducing key HIV infection risk-related behaviors. However, there was considerable clinical and methodological heterogeneity among the trials. Exploratory meta-analysis showed a statistically significant reduction in behaviors associated with high risk of HIV transmission (risk ratio 0.75, 95% confidence interval 0.62 to 0.91). Evidence regarding the effectiveness of behavior change interventions suggests that they are effective in changing behavior associated with HIV transmission. Exploratory stratified meta-analyzes suggested that interventions should be delivered face to face and immediately after testing. There are uncertainties around the generalisability of these findings to the UK setting. please cite doi:10.3310/hta21050.
Methods
- adopt the prisma and picots framework to improve the research protocol
- add the flow diagram of the prisma model
Results
- in figure 1 add the p value of the group comparison
- Add p value comparison in table II
Author Response
- Line 42 – We recognise that the Covid-19 pandemic has had an impact on the general population at different levels. However, this paragraph specifically suggests that the overall reduction in STI diagnoses observed during 2020 may be a reflection of the barriers to access sexual health services as opposed to a genuine reduction in STI rates. The article suggested by the reviewer shows greater stress loads, anxiety and reduced work performance and sleep in a group of health workers in an Italian hospital as a result of wearing masks, we do not think this is of direct relevant to this paper.
- We have referenced the article suggested by the reviewer “Overall, systematic reviews of individual-level behaviour change interventions have demonstrated their effectiveness in reducing key infection risk-related behaviours, although more research is needed to determine the generalisability of these results to the UK setting [15]” (line 93)
- We are grateful to the reviewer for the suggestion and we will consider it for our future research publications. However, we believe that the Pico and Prisma frameworks are better suited to report clinical trials or systematic reviews and meta-analyses.
- Table 2 provides demographic information about the sample; we don’t think that including inferential statistics at this point would be helpful. We are a bit confused by the suggestion of adding p values for group comparison in Figure 1. There were no group comparisons reported in this paper. The one-sample t-tests reported in lines 296-297 assessed participants’ response against the neutral rating in the likert scale, showing that participants reported a significant change in their attitudes after seeing both campaigns. Figure 1 shows that the campaign “Come together safely this summer” had a greater percentage of responses from participants in the "I would book/order an STI test” and “I would buy/order condoms” questions, which helped decision makers ultimately select this campaign.
Reviewer 2 Report
Thank you very much for sending me this manuscript. This paper is very interesting and important. I recommend that this paper should be published. I have several very minor comments for the authors.
- I see the variables were collected on gender, age, and sexual orientation. Were you able to capture any information on immigrant status and racial minority status? These variables are important to capture the cultural construct of sexual behaviours. If not collected, would you discuss this as part of the limitation?
- Another limitation is that there is a selection bias into attending to this type of public health campaign. In particular, there are a group of healthcare professionals who are interested in this type of campaign and those are not. In this study, the voice of the former is well captured but not that of the latter.
- I also question the role of structural problems in this issue. I value and respect the role of behavioural change paradigm. However, healthcare professionals need to understand that youth’s sexual behaviours are heavily influenced by deep-rooted causes such as poverty and discrimination. How does behavioural change paradigm can add to this situation?
Author Response
- We thank the reviewer for this suggestion. We did not gather data on immigrant or racial minority status but we agree that this could have been informative. We have now acknowledged this in the limitation section. (line 359) “A limitation of the current study is that it did not capture information on immigration, socio-economic status or ethnicity of the participants involved. Whilst individual-level behaviour change interventions have been shown to be effective against infection risk-related behaviours, it is essential to acknowledge the influence that broader structural factors, such as inequality, have on sexual health behaviour. Capturing these data would be important to create more effective targeted campaigns and services in the future.”
- We thank the reviewer for this observation. The workshop included sexual health experts because they were part of the NHS trust commissioning the campaign. However, it would be interesting to include non-expert opinions in future studies.
- Whilst we agree that structural factors such as socio-economic status and discrimination influence healthcare behaviours and outcomes, and that this should be taken into account by healthcare professionals, this campaign was limited in scope and could only address individual-level behaviour change, which has been shown to effectively reduced risk behaviours. We have now referenced the role of structural factors in the limitations (line 359)